# Understanding and Steering the Cognitive Behaviors of Reasoning Models at Test-Time

## Abstract

Large Language Models (LLMs) often rely on long chain-of-thought (CoT) reasoning to solve complex tasks. While effective, these trajectories are frequently inefficient—leading to high latency from excessive token generation, or unstable reasoning that alternates between underthinking (shallow, inconsistent steps) and overthinking (repetitive, verbose reasoning). In this work, we study the structure of reasoning trajectories and uncover specialized attention heads that correlate with distinct cognitive behaviors such as verification and backtracking. By lightly intervening on these heads at inference time, we can steer the model away from inefficient modes. Building on this insight, we propose CREST—a training-free method for Cognitive REasoning Steering at Test-time. CREST has two components: (1) an offline calibration step that identifies cognitive heads and derives head-specific steering vectors, and (2) an inference-time procedure that rotates hidden representations to suppress components along those vectors. CREST adaptively suppresses unproductive reasoning behaviors, yielding both higher accuracy and lower computational cost. Across diverse reasoning benchmarks and models, CREST improves accuracy by up to 17.5% while reducing token usage by 37.6%, offering a simple and effective pathway to faster, more reliable LLM reasoning. Code will be made public upon acceptance.

## 1 Introduction

Recent advances in Reinforcement Learning (RL)-based training (Shao et al., 2024) have substantially improved the reasoning capabilities of large language models (LLMs), enabling the emergence of "aha" moments and allowing them to excel in complex tasks such as coding (Jiang et al., 2024), mathematical theorem proving (Shao et al., 2024; Xin et al., 2024), and planning (Huang et al., 2024; Valmeekam et al., 2023). This capability is largely enabled by extended Chain-of-Thought (CoT) reasoning processes. While effective, the reasoning trajectories generated by LLMs are often suboptimal. From an efficiency perspective, long CoT processes consume significantly more tokens than standard responses, leading to increased latency, especially problematic for on-device applications. In terms of performance, recent studies have shown that LLMs often struggle with overthinking (Chen et al., 2024), generating unnecessarily verbose explanations for simple problems, and underthinking (Wang et al., 2025), where they halt reasoning prematurely before fully exploring complex solutions. Surprisingly, some work even suggests that effective reasoning can emerge without any explicit thinking process (Ma et al., 2025a).

To guide and enhance the reasoning process, prior work has primarily focused on directly controlling response length (Muennighoff et al., 2025; Luo et al., 2025a; Ma et al., 2025b; Sun et al., 2025; Yang et al., 2025c). However, there has been limited exploration of the internal cognitive mechanisms that underlie and drive these reasoning behaviors. Drawing inspiration from cognitive psychology, where deliberate processes such as planning, verification, and backtracking, often associated with System 2 thinking, are known to enhance human problem-solving, we posit that analogous cognitive behaviors can be identified and, importantly, steered within LLMs. In particular, we hypothesize that certain components of the model, such as attention heads, specialize in tracking and modulating these distinct reasoning patterns.

In this work, we categorize reasoning processes into two types: linear reasoning (i.e., step-by-step problem solving) and non-linear reasoning (e.g., backtracking, verification, and other divergent

behaviors (Gandhi et al., 2025)). To understand how these behaviors are represented in the activation space, we label individual reasoning steps accordingly and train a simple linear classifier to distinguish between them based on hidden activations. Using linear probes, we identify a small subset of attention heads, referred to as cognitive heads, whose activations are highly predictive of reasoning type. By intervening on these heads during inference, we can steer the model's cognitive trajectory without additional training, reducing redundant steps or encouraging deeper reasoning as needed.

Based on these findings, we introduce CREST (**C**ognitive **RE**asoning **S**teering at **T**est-time), a training-free framework for dynamically adjusting reasoning behaviors during inference. CREST operates by first performing a simple offline calibration to identify cognitive heads and compute steering vectors from representative reasoning examples. Then, during test-time, it uses activation interventions based on these vectors to adaptively guide the model's reasoning trajectory, suppressing inefficient cognitive modes and encouraging effective reasoning behavior. Importantly, CREST is compatible with a wide range of pre-trained LLMs and does not require any task-specific retraining or gradient updates, making it highly scalable and practical for real-world applications. And the test-time steering incurs negligible overhead, achieving matching throughput while reducing token consumption, thereby leading to an overall end-to-end efficiency gain.

In summary, our key contributions are as follows: (i) **Cognitive Head Discovery**: We provide empirical evidence for the existence of cognitive attention heads that correlate with specific reasoning behaviors, offering new interpretability into how cognitive patterns are represented within a model's hidden states. (ii) **Test-Time Behavioral Steering**: We propose a plug-and-play activation intervention technique that enables test-time steering of reasoning behaviors without additional training. (iii) **Comprehensive Evaluation**: We validate our method across a diverse reasoning benchmarks, including MATH500, AMC23, AIME, LiveCodeBench, GPQA-D and Calender Planning, demonstrating that CREST not only enhances reasoning accuracy (up to 17.50%, R1-1.5B on AMC23) but also substantially reduces token usage (up to 37.60%, R1-1.5B on AMC23).

## 2 RELATED WORKS

We organized prior research into three categories and move more related works in Appendix A.

**Reasoning Models.** Early chain-of-thought (CoT) prompting (Wei et al., 2022) and self-consistency decoding (Wang et al., 2022) demonstrated that sampling diverse reasoning paths and selecting the majority answer improves accuracy. Structured search frameworks extend this idea: Tree-of-Thought (Yao, 2023), Graph-of-Thought (Besta et al., 2024), and Forest-of-Thought (Bi et al., 2024). Recent "thinking" model releases include OpenAI's *o*-series (Jaech et al., 2024), Anthropic's *Claude-3.7-Sonnet-Thinking* (Anthropic, 2025), and Google's *Gemini-2.5-Flash* (Google, 2025), alongside competitive open-source models such as DeepSeek-R1 (Guo et al., 2025), Phi-4-Reasoning (Abdin et al., 2025), and Qwen3 (Team, 2025b). These advances enhance models' reasoning abilities and create new possibilities for in-depth analysis of their internal mechanisms.

**Cognitive Behaviors in LLMs.** Recent work defines *cognitive behaviors* as recurring patterns in reasoning traces—such as verification, backtracking, or sub-goal planning—that correlate with accuracy (Gandhi et al., 2025). These mirror human problem-solving heuristics (Newell & Simon, 1972; Gick & Holyoak, 1980; Koriat, 2012; Toth & Campbell, 2022) and motivate methods that explicitly instill similar behaviors in LLMs (Wei, 2022; Wang et al., 2022; Yao, 2023). Our work extends this line by identifying internal attention heads linked to such behaviors.

**Improving Test-Time Reasoning.** Inference-time methods enhance reasoning without retraining. Notable approaches include: (i) adaptive compute control, which dynamically allocates tokens (Han et al., 2025; Xiao et al., 2025), and (ii) direct trace manipulation, which edits or compresses chains-of-thought (Xu et al., 2025b; Cui et al., 2025). More recently, activation editing methods steer hidden representations directly (Turner et al., 2024; Zou et al., 2025; Huang et al., 2025). Our approach, CREST, advances this strand by identifying *cognitive attention heads* and demonstrating targeted head-level interventions that improve efficiency while providing new interpretability insights.

## 3 DISSECTING AND MODULATING COGNITIVE PATTERNS IN REASONING

In this section, we examine how reasoning models exhibit and internalize cognitive behaviors, with a particular focus on non-linear thinking patterns such as verification, subgoal formation, and

backtracking. We begin in Section 3.1 by identifying and categorizing these behaviors at the level of individual reasoning steps. Section 3.2 then investigates how such behaviors are reflected in the internal activations of attention heads, revealing a subset, namely, *cognitive heads* that reliably encode non-linear reasoning. Finally, in Section 3.3, we demonstrate that these heads can be directly manipulated at test time to steer the model's reasoning trajectory, offering a mechanism for fine-grained control over complex reasoning without retraining.

## 3.1 COGNITIVE BEHAVIORS IN REASONING MODELS

O1-like LLMs solve problems through extended chain-of-thought reasoning, often exhibiting non-linear patterns that diverge from traditional step-by-step reasoning. These non-linear trajectories (*e.g.,* backtracking, verification, subgoal setting and backward chaining) closely mirror human cognitive behaviors and enhance the model's ability to tackle complex problem-solving tasks (Gandhi et al., 2025). To analyze cognitive behaviors, we segment the reasoning process, which is typically bounded by the `<think>` and `</think>` markers tokens into discrete reasoning steps, each delimited by the token sequence "$\backslash n \backslash n$". We then categorize each reasoning step into one of two types using keyword matching: **Non-linear Reasoning**, if the reasoning step contains any keyword from a predefined set (*e.g.,* {Wait, Alternatively}; full list in Appendix B.1), it is labeled as non-linear; otherwise, it is classified as a **Linear Reasoning** step. We denote a single reasoning step, composed of multiple tokens, as S, and use $S^l$ and $S^n$ to represent linear and non-linear reasoning steps, respectively.

## 3.2 IDENTIFYING ATTENTION HEADS OF COGNITIVE BEHAVIORS

Analyzing cognitive behaviors during reasoning is inherently challenging, as for the same behavior, such as verification, can manifest differently across the token space, depending on the sample's context and the underlying reasoning pattern. Intuitively, these behaviors often involve long-range token interactions, where the model retrieves and re-evaluates previous reasoning steps. Meanwhile, recent studies (Olsson et al., 2022; Elhage et al., 2021; Wu et al., 2024) have shown that attention heads frequently perform distinct and interpretable functions, such as tracking, factual retrieval, and position alignment. This points toward a modular architecture in which specific heads may specialize in different cognitive sub-tasks. Motivated by this insight, we conduct a preliminary study and identify attention heads that are strongly correlated with cognitive behaviors during reasoning.

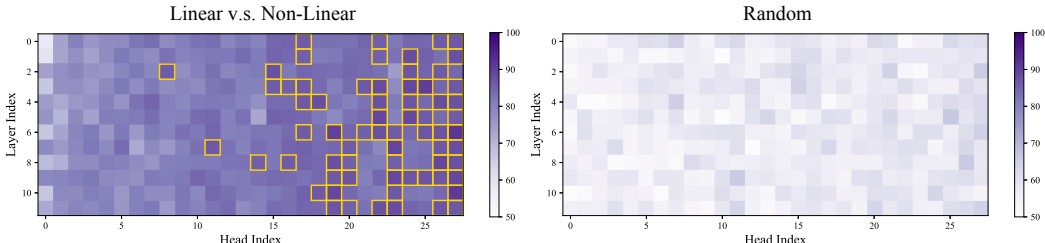

Figure 1: Visualization of probing accuracy for DeepSeek-R1-Distill-Qwen-1.5B. (Left) Accuracy on linear and non-linear reasoning steps, with high-accuracy regions (*i.e.,* larger than 85%) highlighted in gold boxes. (Right) Accuracy measured across randomly sampled tokens. See **Setup** in Section 3.2.

**Setup.** We begin by randomly sampling 500 training examples from the MATH-500 benchmark (Lightman et al., 2023) and running end-to-end inference with the DeepSeek-R1-Distill-Qwen-1.5B model. Crucially, we define a **"step"** as the contiguous chunk of reasoning text between two occurrences of the special delimiter token $\backslash n \backslash n$.

1. **Segment.** For every prompt, split the chain-of-thought at the delimiter $\backslash n \backslash n$, producing $k$ segments $\{s_1, s_2, \ldots, s_k\}$. Because the delimiter is kept, $\backslash n \backslash n$ is the final token of each segment, so every $s_\ell$ (with $\ell = 1, \ldots, k$) represents one discrete *thinking step*.
2. **Embed each step.** Re-run inference on the chain-of-thought $\{s_1, s_2, ..., s_k\}$ as one single prefill and capture the hidden state at the segment-terminating $\backslash n \backslash n$ token. Treat this vector as a compact summary of the preceding tokens, and extract the post-attention activations

$$a_{s_k}^{i,j} \in \mathbb{R}^d, \qquad i = 1 \ldots H, \ j = 1 \ldots L, \tag{1}$$

where $i$ indexes heads and $j$ layers. Thus, $a_{s_k}^{i,j}$ represents the contextual embedding of the delimiter token (\n\n) at the end of segment $s_k$.

3. **Label & probe.** Mark each step as *linear* ($y_{s_k} = 0$) or *non-linear* ($y_{s_k} = 1$). For every head $(i, j)$ fit a linear probe $\theta^{i,j} = \arg\min_\theta \mathbb{E}\left[ f\left( y_{s_k}, \sigma(\theta^\top a_{s_k}^{i,j}) \right) \right]$, where $\sigma$ is the sigmoid and $f$ is mean-squared error loss function. See the training details in Appendix.

The resulting probes pinpoint heads whose activations best distinguish linear from non-linear reasoning and supply the foundation for the calibration and steering stages that follow.

Across multiple prompts. For each prompt $\ell$, segmentation yields $k_\ell$ steps $S^{(\ell)} = \{s_1^{(\ell)}, \ldots, s_{k_\ell}^{(\ell)}\}$. Collectively these form the global set $S = \bigcup_{\ell=1}^n S^{(\ell)}$, whose size is $|S| = \sum_{\ell=1}^n k_\ell$. Every $S^{(\ell)} \in S$ is embedded, labeled, and probed exactly as described above, so all downstream analyses operate on the full collection of $\sum_{\ell=1}^n k_\ell$ reasoning segments. We define $a_{s_k^{(\ell)}}^{i,j}$ for prompt $\ell$.

**Results**. The classification accuracy is shown in Figure 1, with additional results across different models and datasets provided in Appendix C.1. As a sanity check, we repeat the probing procedure on randomly sampled tokens, shown in the right part of Figure 1, where the classification accuracy remains near chance level—indicating no distinguishable signal. In contrast, the left subfigure reveals that certain attention heads achieve significantly higher accuracy. We refer to these as **Cognitive Heads**, while the remaining are treated as standard heads. Notably, cognitive heads are more prevalent in deeper layers, which is aligned with the expectation that deeper layers capture higher-level semantic features and shallow layers encode token-level features (Ethayarajh, 2019; Liu et al., 2019). Some cognitive heads also emerge in middle layers, suggesting a distributed emergence of cognitive functionality across the model.

### 3.3 Manipulating Cognitive Behaviors via Activation Intervention

We then investigate whether nonlinear chains of thought can be modulated *at test time* by directly editing the activations of the most "cognitive" attention heads, following the methodology of (Sun et al., 2025).

**Prototype construction.** With the definition in **Setup**. For a prompt, we have $N_\ell = \sum_{k=1}^{|S^{(\ell)}|} \mathbb{I}[y_{s_k^{(\ell)}} = 1]$ non-linear thoughts. With $v_\ell^{i,j} = \frac{1}{N_\ell} \sum_{k=1}^{|S^{(\ell)}|} a_{s_k^{(\ell)}}^{i,j} \mathbb{I}[y_{s_k^{(\ell)}} = 1]$ defined as non-linear average activation for $\ell$-th prompt, we form a head-specific vector capturing the average pattern of nonlinear reasoning:

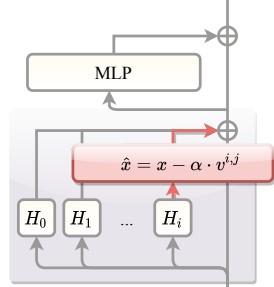

Figure 2: Illustration of cognitive reasoning steering at test-time.

$$v^{i,j} = \frac{1}{N} \sum_{\ell=1}^n N_\ell v_\ell^{i,j} \quad \text{with} \quad N = \sum_{\ell=1}^n N_\ell, \quad (2)$$

Thus, $v^{i,j}$ represents the mean activation across all non-linear steps.

**Online intervention.** As shown in Figure 2, we pause after each reasoning step (i.e., after generating \n\n), select the top 7% of attention heads (ranked by the classification-accuracy metric in equation 3), and modify their activations via

$$\hat{x}^{i,j} = x^{i,j} - \alpha \, v^{i,j} \quad (3)$$

Here, $\alpha$ is a tunable scalar controlling intervention strength: $\alpha > 0$ attenuates nonlinear behavior, while $\alpha < 0$ amplifies it. Notably, $x^{i,j}$ corresponds to the post-attention state at inference, whereas $v^{i,j}$ summarizes activation at \n\n positions.

As shown in Figure 3, we pause the reasoning process at Step 9, during which all previous steps followed a linear reasoning trajectory. In the original process, the subsequent step initiates a non-linear reasoning pattern—specifically, a backward chaining behavior (Gandhi et al., 2025)—starting with the word "alternatively." However, after applying activation intervention to suppress non-linear reasoning, the model continues along a linear trajectory and still arrives at the correct final answer. Conversely, we pause the model at Step 10—after it completes a non-linear segment and resumes linear reasoning. In this case, we enhance the non-linear component via activation intervention, causing the model to continue along a non-linear path instead.

**Question:** Convert the point $(0, 3)$ in rectangular coordinates to polar coordinates. Enter your answer in the form $(r, \theta)$, where $r > 0$ and $0 \le \theta < 2\pi$

*Original Reasoning Process*

Okay, so I have this problem where I need to convert the rectangular coordinates $(0, 3)$ to polar coordinates.
......
I remember that when x is zero, and y is positive, θ is π/2. Similarly, when x is zero and y is negative, θ is 3π/2. So, in this case, since y is positive, θ is π/2.

**Alternatively, sometimes when dealing with arctangent,** if x is zero, we have to consider the angle. So, arctan(y/x) where x is zero would be undefined, but in this case, since x is zero and y is positive, the angle is π/2.
......
Final Answer The polar coordinates are $\boxed{\left(3, \frac{\pi}{2}\right)}$. </think>

*Reduce Non-Linear Reasoning*

So, putting it all together, the polar coordinates for the point $(0, 3)$ are $(3, \pi/2)$. 1. Calculated r: $\sqrt{(0^2 + 3^2)} = 3$. That seems correct. 2. Determined θ: Since x is 0 and y is positive, θ is π/2. Correct. I think that's right. So, I'm confident that $(0, 3)$ in rectangular coordinates is $(3, \pi/2)$ in polar coordinates.
Final Answer The polar coordinates are $\boxed{\left(3, \frac{\pi}{2}\right)}$ </think>

**But hold on, let me double-check. Sometimes,** depending on the quadrant, but in this case, since x is 0 and y is positive, it's on the positive y-axis, which is the boundary between the first and second quadrants. But in this case, it's just the positive y-axis. So, the angle is definitely π/2.
**Wait, another thought: when I use arctan(y/x),** ....
......
Final Answer The polar coordinates are $\boxed{\left(3, \frac{\pi}{2}\right)}$ </think>

Enhance Non-Linear Reasoning

Figure 3: A detailed example illustrating how intervening in the original reasoning by enhancing or suppressing non-linear reasoning, alters the reasoning trajectory. Results are obtained using DeepSeek-R1-Distill-Qwen-1.5B. Please zoom in for clearer visualization.

While all versions of the reasoning process ultimately produce the correct final answer, they differ significantly in trajectory length: the original process takes 17 steps, the reduced non-linear path takes only 12 steps, and the enhanced non-linear path extends to 45 steps, implying potential redundancy in current reasoning processes. To further quantify the effects of the intervention, we collect statistical results from the intervention process. Using 100 samples from the MATH500 test set, we observe that the DeepSeek-R1-Distill-Qwen-1.5B model takes an average of 22.83 steps to complete the reasoning process. When varying the intervention strength, the number of non-linear reasoning steps adjusts accordingly. In con-

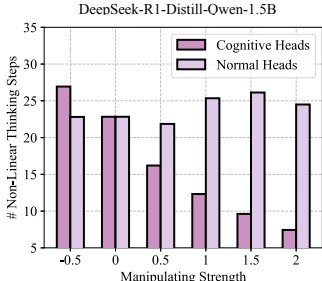

Figure 4: Statistical analysis of the number of reasoning steps under varying levels of intervention strength $\alpha$ in equation 3.

trast, when applying the same manipulation to non-cognitive (*i.e.*, normal) heads—specifically, the bottom 7% of attention heads with the lowest classification accuracy—the number of reasoning steps remains largely unchanged across different intervention strengths, as shown in Figure 4. These results support the existence of cognitive attention heads and demonstrate the feasibility of manipulating cognitive behaviors during reasoning.

## 4 CREST: COGNITIVE REASONING STEERING AT TEST-TIME

As observed in the previous section, the model is able to arrive at the correct final answer with fewer non-linear reasoning steps, suggesting the presence of redundant reasoning that hinders end-to-end efficiency. Motivated by these insights, we propose a training-free strategy to adaptively adjust the reasoning process during inference. Our framework consists of two main processes: an offline calibration stage, along with a test-time steering stage.

### 4.1 OFFLINE CALIBRATION

We perform the following two steps to process the head vectors for controlling the reasoning process. It is worth noting that this offline calibration stage is a one-shot procedure, requiring only negligible cost compared to LLM training and incurring no additional latency during subsequent inference.

#### 4.1.1 IDENTIFYING COGNITIVE HEADS.

We begin by locating the *cognitive* attention heads that matter most for reasoning, details as follows:

1. **Calibration dataset and Probing.** As describe in **Setup** of Section 3.2, we draw some training samples, embed each step, labeled, and probe to every attention head and rank them by accuracy.

2. **Selection.** Keep the top $10\%$ of heads. For each retained head $(i,j)$, we pre-compute $v^{i,j}$ as defined in 3.3, the average hidden state across the non-linear reasoning steps.

### 4.1.2 Aligning head-specific vectors via low-rank projection.

Since the head vector is derived from a specific calibration dataset and identified through keyword matching to capture non-linear reasoning steps, it inevitably carries noise within the activation space. As a result, the head-specific vector becomes entangled with irrelevant components and can be expressed as

$$v^{i,j} = v^{i,j}_{\text{reason}} + v^{i,j}_{\text{noise}},$$

where $v^{i,j}_{\text{reason}}$ denotes the true non-linear reasoning direction, and $v^{i,j}_{\text{noise}}$ represents spurious components. This concern is further supported by recent findings that length-aware activation directions can also be noisy (Huang et al., 2025).

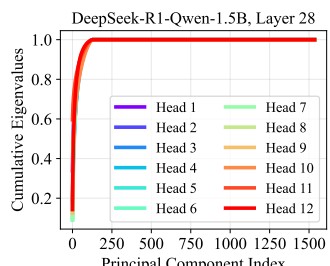

Figure 5: Cumulative Eigenvalues of the covariance matrix of head vectors in the last layer of DeepSeek-R1-Qwen-1.5B. The PCA matrix $A$ here is of dimension $d \times d$. Notably, the top $100$ principal components already capture nearly all of the variance, indicating that the effective dimensionality of the head activations is much lower than the raw space.

To address this, we analyze the covariance structure of the collected activations. Specifically, given a set of activations $\{a^{i,j}_{s_k}\}$, we concatenate activations from all steps into a single matrix: $A^{i,j} = \left[a^{i,j}_{s_k}\right] \in \mathbb{R}^{d \times N}$. We compute the empirical covariance matrix and perform its eigen-decomposition as follows:

$$\Sigma^{i,j} \;=\; \frac{1}{N}\sum_{k=1}^{N}\big(A^{i,j}_k - \bar{A}^{i,j}\big)\big(A^{i,j}_k - \bar{A}^{i,j}\big)^{\top}; \Sigma^{i,j} \;=\; Q^{i,j}\Lambda^{i,j}\big(Q^{i,j}\big)^{\top} \tag{4}$$

where $\bar{A}^{i,j}$ is the average activation across $N$ samples. We then visualize the distribution of cumulative eigenvalues, as shown in Figure 5.

We observe that the signal-to-noise ratio of the raw head vector is low, with the critical information concentrated in a low-rank subspace. To remove such redundancy, we perform a low-rank projection to constrain the head vector into an informative subspace. However, if each head is assigned its own subspace, the resulting representations may lose comparability across heads, as the shared space is replaced by distinct, head-specific subspaces. Therefore, we adopt a shared subspace to filter out the noise components of head vectors. Instead of computing the head-specific covariance matrix $\Sigma^{i,j}$, we aggregate the activations of all heads within a layer, $A^j = \left[\sum_{i=1}^{N_h} a^{i,j}_{s_k}\right] \in \mathbb{R}^{d \times N}$, where $N_h$ is the number of heads in layer $j$ and $N$ is the number of samples. We then compute the eigenspace $Q^j$ from the covariance of $A^j$, and project each head vector $v^{i,j}$ onto the top-$n$ eigenvectors to obtain the aligned representation:

$$\hat{v}^{i,j} \;=\; Q^j[:,:n]\, Q^j[:,:n]^{\top} v^{i,j}$$

### 4.2 Test-time Steering

During decoding, immediately after each reasoning step, we rotate the representation of the last token to enforce orthogonality with the pre-computed steering direction, while preserving the original activation magnitude:

$$\hat{x}^{i,j} = \frac{\|x^{i,j}\|}{\|x^{i,j} - \big((x^{i,j})^{\top}v^{i,j}\big)v^{i,j}\|} \left(x^{i,j} - \big((x^{i,j})^{\top}v^{i,j}\big)v^{i,j}\right), \tag{5}$$

where $x^{i,j}$ denotes the original representation and $v^{i,j}$ is the steering direction. We use $\ell_2$ norm here.

The main motivation behind this design is to eliminate the dependence on hyperparameters. Previous steering methods require tuning the steering strength for each model (Huang et al., 2025; Chen et al., 2025), which limits their practical applicability due to the need for careful hyperparameter adjustment. In contrast, by preserving the activation norm, we avoid the need for such tuning.

Moreover, activation outliers are a well-known issue in LLMs, often leading to highly unstable activation magnitudes (Sun et al., 2024; Nrusimha et al., 2024). Our norm-preserving strategy mitigates this problem by preventing large norm fluctuations during inference, thereby making the steering process more stable.

## 5 EXPERIMENTS

### 5.1 IMPLEMENTATION DETAILS

**Models & Datasets.** We conduct experiments on widely used reasoning models of different scales, including DeepSeek-R1-Distill-Qwen-1.5B/7B/32B (R1-1.5B/7B/32B) (Guo et al., 2025), Qwen3-4B/30B (Yang et al., 2025a), and GPT-OSS-20B (Agarwal et al., 2025). Evaluation is performed across a diverse set of reasoning benchmarks: MATH500 (Hendrycks et al., 2021; HuggingFaceH4, 2024), LiveCodeBench (Jain et al., 2024), AIME (Patel et al., 2024) (120 problems from the 2022–2025 American Invitational Mathematics Examination), AMC23 (math ai, 2023), GPQA-D (Rein et al., 2024), and Calendar Planning (Zheng et al., 2024).

**Baselines.** We compare CREST against training-free methods and include four competitive baselines from diverse perspectives: (i) Thought Switching Penalty (TIP) (Wang et al., 2025), which suppresses the logits of specific tokens (e.g., "Alternatively," "Wait") to reduce unnecessary shifts in reasoning trajectories; (ii) SEAL (Chen et al., 2025), which performs task arithmetic in the latent space to down-regulate internal representations associated with such tokens; (iii) Dynasor (Fu et al., 2024), which reduces token cost by performing early exit based on a consistency criterion during decoding; and (iv) Soft-Thinking (Zhang et al., 2025), which enables latent-space reasoning with an entropy-based early-exit strategy. In addition, we include the original full model as a baseline (Vanilla).

**Hyperparameters.** In CREST, the only hyperparameter is the number of attention heads to steer. To avoid task-specific tuning, we conduct a preliminary ablation study in Section 5.3.1 and fix this setting for each model across all tasks. During decoding, we use the default settings: temperature = 0.6, top-p = 0.95, and a maximum generation length of 32,768 tokens.

### 5.2 TOKEN-EFFICIENT REASONING WITH SUPERIOR PERFORMANCE

Table 1: Comparison results against other baselines across various tasks. Note that CREST employs consistent head vectors and a fixed number of steered heads for all tasks, avoiding task-specific hyperparameter tuning.

| Model | Methods | MATH500 | | AIME25 | | AIME22-24 | | AMC23 | |
|---|---|---|---|---|---|---|---|---|---|
| | | Pass@1 (↑) | #Tokens (↓) | Pass@1 (↑) | #Tokens (↓) | Pass@1 (↑) | #Tokens (↓) | Pass@1 (↑) | #Tokens (↓) |
| R1-1.5B | Vanilla | 84.00 | 5497 | 20.00 | 15974 | 17.80 | 17034 | 72.50 | 8951 |
| | TIP | 83.40 | 4414 | 20.00 | 14200 | **24.40** | 14157 | 72.50 | 8069 |
| | SEAL | 81.60 | 4150 | 16.70 | 17153 | 22.20 | 14207 | 67.50 | 8202 |
| | Dynasor | 89.00 | **3267** | 28.00 | 12412 | 24.12 | 15337 | 70.00 | 7782 |
| | Soft-Thinking | 66.80 | 9401 | 23.30 | 14843 | 12.20 | 18418 | 55.00 | 13160 |
| | CREST | 84.80 | 4106 | **30.00** | **11101** | 20.00 | **13388** | **90.00** | **5584** |
| | *% Gain from Vanilla* | *0.8%* | *25.3%* | *10.0%* | *30.5%* | *2.2%* | *21.4%* | *17.5%* | *37.6%* |
| R1-7B | Vanilla | 91.60 | 4020 | 43.33 | 12139 | 44.40 | 13709 | 87.50 | 5912 |
| | TIP | **92.40** | 3173 | 33.30 | 11225 | 44.40 | 11112 | 90.00 | 5532 |
| | SEAL | 91.20 | 3335 | 36.70 | 11692 | 42.22 | 12448 | 87.50 | 4784 |
| | Dynasor | 92.00 | 3619 | 41.00 | 9360 | 45.10 | 10314 | 75.00 | 7809 |
| | Soft-Thinking | 90.00 | 4095 | 33.30 | 11370 | 35.60 | 12551 | 80.00 | 5859 |
| | CREST | **92.40** | **2661** | **43.33** | **8083** | **44.40** | **9488** | **92.50** | **3937** |
| | *% Gain from Vanilla* | *0.8%* | *33.8%* | *0.0%* | *33.4%* | *0.0%* | *30.8%* | *5.0%* | *33.4%* |

**Superior Performance against Other Baselines**. To begin, we demonstrate that CREST can reduce the token cost while achieving superior performance. As shown in Table 1, on R1-1.5B, CREST consistently improves over the vanilla baseline. For instance, on AMC23, CREST attains 90.% Pass@1 while lowering the average token cost from 8951 to 5584, a substantial 37.6% reduction. The trend persists at larger model scales. With R1-7B, CREST achieves 92.4% accuracy on MATH500 with only 2661 tokens, representing a 34% cost reduction compared to vanilla, while exceeding other competitive baselines such as TIP and Dynasor. Overall, these results highlight the strength of CREST in jointly optimizing accuracy and efficiency. Unlike prior baselines, which often trade one for the other, CREST consistently demonstrates gains across both metrics, validating its generality.

**Consistent Improvements Across Model Sizes and Architectures**. As shown in Table 2, we further evaluate CREST across a wide range of model sizes, from 1.5B to 32B, and across different

architectures, including Qwen-2, Qwen-3, and GPT-OSS. In each subfigure, the token reduction ratio is visualized with horizontal arrows, while the accuracy improvements are indicated by vertical arrows. The results demonstrate that CREST consistently benefits diverse model families. In some cases, the token reduction ratio reaches as high as 30.8% (R1-7B on AIME22-24), while the accuracy improvement peaks at 6.7% (GPT-OSS-20B on AIME25). These findings provide strong evidence of the generalization ability of CREST across both model scales and architectures.

Table 2: CREST demonstrates generalization across diverse model architectures, from dense models (R1-1.5B, R1-7B, R1-32B) to mixture-of-experts models (GPT-OSS-20B, Qwen3-30B). Arrows indicate the transition from Vanilla → CREST, and ∆Tok denotes the percentage reduction in average tokens (context length).

| Model | AIME2025 | | | | AIME22–24 | | | |
|---|---|---|---|---|---|---|---|---|
| | Acc (V→C) | ∆Acc | Tokens (V→C) | ∆Token | Acc (V→C) | ∆Acc | Tokens (V→C) | ∆Token |
| R1-1.5B | 17.0 → 20.3 | ↑ 3.3% | 15,986 → 12,393 | ↓ 22.5% | 18.0 → 20.2 | ↑ 2.2% | 17,052 → 13,407 | ↓ 21.4% |
| R1-7B | 43.5 → 43.5 | ↑ 0.0% | 12,114 → 8,058 | ↓ 33.4% | 44.0 → 44.0 | ↑ 0.0% | 13,692 → 9,471 | ↓ 30.8% |
| R1-32B | 57.7 → 61.0 | ↑ 3.3% | 12,747 → 10,274 | ↓ 19.4% | 64.0 → 64.0 | ↑ 0.0% | 11,465 → 9,730 | ↓ 15.1% |
| GPT-OSS-20B | 50.0 → 56.7 | ↑ 6.7% | 22,930 → 17,665 | ↓ 22.4% | 60.0 → 62.0 | ↑ 2.0% | 22,207 → 20,455 | ↓ 7.9% |
| Qwen3-30B | 73.30 → 73.33 | ↑ 0.03% | 15,936 → 14,568 | ↓ 8.6% | 78.0 → 78.0 | ↑ 0.0% | 15,292 → 13,973 | ↓ 8.6% |

Table 3: Comparison results against other baselines across various tasks. Note that CREST employs consistent head vectors and a fixed number of steered heads for all tasks, avoiding task-specific hyperparameter tuning.

| Model | Methods | AIME22-25 (Math) | | LiveCodeBench (Code) | | GPQA-D (Common-Sense) | | Calendar Planning (Plan) | |
|---|---|---|---|---|---|---|---|---|---|
| | | Pass@1 (↑) | #Tokens (↓) | Pass@1 (↑) | #Tokens (↓) | Pass@1 (↑) | #Tokens (↓) | Pass@1 (↑) | #Tokens (↓) |
| R1-32B | Vanilla | 62.18 | 11823 | 56.29 | 10830 | 32.32 | 7600 | 77.10 | 3145 |
| | CREST | 63.00 | 9903 | 59.28 | 9541 | 40.91 | 6627 | 78.70 | 2507 |
| | % Gain | 1.3% | 16.2% | 5.3% | 11.9% | 26.6% | 12.8% | 2.1% | 20.3% |
| Qwen3-30B | Vanilla | 77.49 | 15456 | 66.47 | 15307 | 70.20 | 7013 | 66.20 | 5869 |
| | CREST | 77.50 | 14135 | 73.05 | 15317 | 70.20 | 6592 | 68.10 | 5767 |
| | % Gain | 0.01% | 8.5% | 9.9% | -0.07% | 0.0% | 6.0% | 2.9% | 1.7% |

**Strong Generalization Across Diverse Task Domains.** We further evaluate CREST across multiple task domains, including mathematical reasoning (AIME22–25, comprising all 120 problems from 2022–2025), code generation (LiveCodeBench), common-sense reasoning (GPQA-D), and planning (Calendar Planning), as reported in Table 3. Despite being calibrated only on MATH500, CREST generalizes effectively to both in-domain and out-of-domain tasks. Within the math domain, it maintains strong transfer, achieving 63.% accuracy on AIME22–25 while reducing token cost from 11,823 to 9,903. Beyond math, CREST delivers consistent improvements: on LiveCodeBench, accuracy increases from 56.3% to 59.3% with fewer tokens; on GPQA-D, accuracy rises substantially from 32.3% to 40.9% while tokens drop from 7,600 to 6,627; and on Calendar Planning, performance improves from 77.1% to 78.7% with notable cost reduction (3,145 → 2,507). Similar patterns hold for larger architectures like Qwen3-30B, where CREST boosts LiveCodeBench accuracy from 66.5% to 73.1% while also reducing tokens.

**Analysis.** The performance gains of CREST can be largely attributed to the intrinsic redundancy in chain-of-thought reasoning, consistent with recent findings that LLMs can often achieve competitive or even superior performance without explicit reasoning when combined with parallel test-time techniques such as majority voting (Ma et al., 2025a), and that pruning or token-budget-aware strategies applied to reasoning traces do not necessarily harm accuracy (Xia et al., 2025; Luo et al., 2025a). By intervening at the activation level, CREST effectively mitigates this redundancy, achieving a win–win in both efficiency and accuracy.

## 5.3 Further Investigation

### 5.3.1 Ablation Study on the Number of Steered Heads

When implementing CREST, a natural design question concerns the number of attention heads to steer. To investigate this, we conduct ablation studies on R1-1.5B and R1-7B on the AIME22-24 task.

Overall, we find that steering approximately the top 38% of attention heads delivers the strongest performance, balancing both accuracy and token reduction. Figure 6 illustrates the ablation study on the number of attention heads used for intervention. In this analysis, we rank heads by linear probing accuracy and evaluate the top subsets on the AIME22-24 benchmark. The results indicate that steering 38% of all attention heads provides the best balance, yielding improvements in both accuracy and token efficiency.

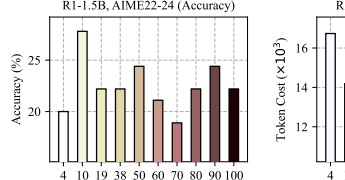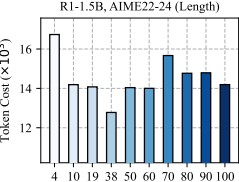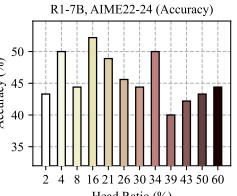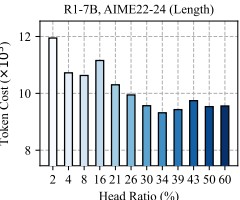

Figure 6: Ablation results on the number of attention heads used for intervention. Darker colors indicate a larger proportion of heads being steered.

Moreover, we observe that the proportion of steerable heads is relatively stable across different models: both R1-1.5B and R1-7B achieve their best performance at similar attention head ratios. This consistency further confirms the robustness of our approach and highlights its ease of hyperparameter tuning. Consequently, we adopt this 'gold ratio' as the default setting in our experiments, thereby avoiding task-specific tuning that could risk information leakage from the test set.

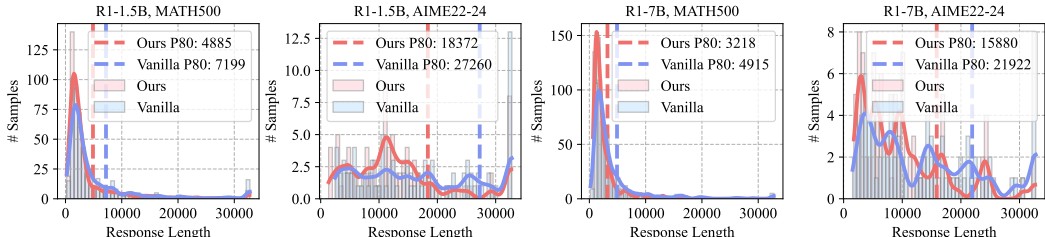

Figure 7: Histogram of Response Lengths. Each subfigure displays the empirical histogram together with a probability density estimate obtained via a Gaussian kernel. The dashed vertical line marks the length threshold covering the top 80% of samples; the corresponding length value is reported in the legend.

### 5.3.2 RESPONSE LENGTH DISTRIBUTION

In Section 5.2, we primarily compared different methods based on the average token cost across the full test set. To gain deeper insights into efficiency improvements, we further analyze the distribution of response lengths. Figure 7 presents histograms comparing our method with vanilla inference. Each subfigure shows both the distribution and the token cost for the top 8% of samples. The results reveal that CREST shifts the distribution leftward, highlighting more pronounced token reductions in terms of both the average and the top-8% subset.

We also observe that, under both CREST and vanilla inference, a small number of failure cases reach the maximum generation limit of 32k tokens. Upon closer inspection, these failures typically involve repetitive outputs. This suggests that CREST could be further enhanced by incorporating early-exit strategies to mitigate repetition. We will explore in the future work.

## 6 CONCLUSION

In this paper, we investigate one of the core capabilities of large language models: reasoning. We conduct a series of empirical studies to better understand the reasoning processes of LLMs and categorize extended chain-of-thought reasoning into two types: linear, step-by-step reasoning and cognitive-style non-linear reasoning. Our findings reveal that certain attention heads are correlated with non-linear cognitive reasoning patterns and can be influenced through activation intervention. Based on these insights, we propose CREST, a training-free approach for steering the reasoning trajectory at test time. Through extensive experiments, we demonstrate that CREST improves both reasoning accuracy and inference efficiency without requiring additional training. Moreover, our method is broadly compatible with a wide range of pre-trained LLMs, highlighting its practical potential for enhancing reasoning models in real-world applications.

## 7 REPRODUCIBILITY STATEMENT

We provide detailed descriptions of the experimental settings, including datasets, model architectures, generation hyperparameters, and evaluation protocols, in the main text. All datasets are publicly accessible, and the code for this work will be released publicly upon acceptance.

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

# A    EXTENDED RELATED WORKS

We organized prior research into three key categories and, to the best of our ability, emphasize the most recent contributions from the extensive body of work.

**Reasoning Models.** Early work on chain-of-thought (CoT) prompting Wei et al. (2022) and self-consistency decoding Wang et al. (2022) showed that sampling diverse reasoning paths at inference time and selecting the most frequent answer markedly improves accuracy. Structured search frameworks subsequently generalise this idea: *Tree-of-Thought* performs look-ahead search over branching "thought" sequences Yao (2023); *Graph-of-Thought* re-uses sub-derivations through a non-linear dependency graph Besta et al. (2024); and *Forest-of-Thought* scales to many sparsely activated trees under larger compute budgets Bi et al. (2024). *Since then, the field of reasoning language models has advanced rapidly, driven in large part by innovations in test-time thinking strategies OpenAI (2024); Snell et al. (2025); Sutskever (2024).* Closed-source providers now offer dedicated "thinking" variants such as OpenAI's o-series Jaech et al. (2024), Anthropic's *Claude-3.7-Sonnet-Thinking* Anthropic (2025), and Google's *Gemini-2.5-Flash* Google (2025). The open-source community has kept pace with competitive models including *DeepSeek-R1* Guo et al. (2025), *Qwen2.5* Yang et al. (2024), *QWQ* Team (2024), *Phi-4-Reasoning* Abdin et al. (2025), and, most recently, *Qwen3* Team (2025b), alongside emerging contenders such as *R-Star* Guan et al. (2025), *Kimi-1.5* Team et al. (2025), *Sky* Team (2025a), and *RedStar* Xu et al. (2025a). These open-weight models enable in-depth analysis of their underlying reasoning mechanisms, offering a unique opportunity to "unblack-box" their cognitive processes. In this work, we explore how manipulating internal components, such as attention heads and hidden states, can influence the model's reasoning behavior.

**Cognitive Behaviors in LLMs.** In Gandhi et al. (2025), a *cognitive behavior* is defined as any readily identifiable pattern in a model's chain-of-thought—such as verification (double-checking work), backtracking (abandoning an unfruitful path), sub-goal setting (planning intermediate steps), or backward chaining (reasoning from goal to premises)—that appears in the text trace and statistically correlates with higher task accuracy or more sample-efficient learning. These behaviors mirror classic findings in human problem solving: means–ends sub-goal analysis Newell & Simon (1972), analogical transfer Gick & Holyoak (1980), metacognitive error monitoring Koriat (2012), and adaptive backtracking during search Toth & Campbell (2022). Modern LLM methods explicitly instate the same heuristics—for example, chain-of-thought prompting Wei (2022) makes the reasoning trace visible, while self-consistency sampling Wang et al. (2022) and Tree-of-Thought search Yao (2023) operationalize backtracking and sub-goal exploration. By situating LLM "cognitive behaviors" within this well-studied human framework, we both ground the terminology and reveal gaps where LLMs still diverges from human cognition, motivating a surge of techniques aimed at "teaching" models to think like human.

**Methods to Improve *Test-Time* Reasoning Models.** Rather than modifying training regimes—e.g. self-fine-tuning Muennighoff et al. (2025) or RL curricula such as *Absolute Zero* Zhao et al. (2025)—we review approaches that act *only at inference*. Adapting (and extending) the taxonomy of Sui et al. (2025), we distinguish four lines of work and situate our own method, CREST, within the emerging fourth category.

- *Light-weight tuning.* Small, targeted weight or prompt updates steer models toward brevity without costly retraining. RL with explicit length penalties (*Concise RL*) and *O1-Pruner* shorten chains-of-thought (CoT) while preserving accuracy Fatemi et al. (2025); Luo et al. (2025b). Model-side tweaks such as *ThinkEdit* and an elastic CoT "knob" expose conciseness or length on demand Sun et al. (2025); Ma et al. (2025c). Together these studies reveal an inverted-U length–accuracy curve Wu et al. (2025) that motivates our desire to *steer* (rather than merely shorten) reasoning traces.
- *Adaptive compute control.* The model spends tokens only when they help. *Token-Budget-Aware Reasoning* predicts a per-question budget Han et al. (2025); confidence-based *Fast–Slow Thinking* routes easy instances through a cheap path Xiao et al. (2025); early-exit policies such as *DEE*, *S-GRPO*, and self-adaptive CoT learning halt generation when marginal utility drops Yang et al. (2025b); Dai et al. (2025); Yang et al. (2025d). Our results show that CREST can *combine* with these token-savers, further reducing budget without extra training.

- *Direct trace manipulation.* These methods edit or reuse the textual CoT itself. *SPGR* keeps only perplexity-critical steps Cui et al. (2025); *Chain-of-Draft* compresses full traces to terse "draft" thoughts at ∼8 % of the tokens Xu et al. (2025b); confidence-weighted self-consistency and *WiSE-FT* ensemble weights cut the number of sampled paths or models needed for robust answers Taubenfeld et al. (2025); Dang et al. (2025). While these techniques operate in token space, ours intervenes *inside* the network, offering an orthogonal lever that can coexist with draft-style pruning.
- *Representation-level activation editing.* A newer strand steers generation by *editing hidden activations* rather than weights or outputs. Early examples include Activation Addition (ActAdd) Turner et al. (2024) and Representation Engineering Zou et al. (2025), which inject global steering vectors into the residual stream; PSA adds differential-privacy guarantees to the same idea Goel et al. (2025).

**CREST** advances *representation-level activation editing* by discovering *cognitive attention heads* aligning with concrete reasoning behaviors and showing that *head-specific interventions* outperform global vectors. Beyond performance, our cognitive-head analysis provides new interpretability evidence that bridges recent attention-head studies Zheng et al. (2025) with activation-editing control.

# B    MORE IMPLEMENTATION DETAILS

## B.1    KEYWORD LIST FOR CATEGORIZING REASONING STEPS

To categorize thinking steps into linear and non-linear reasoning types, we adopt a keyword-matching strategy. Specifically, if a step contains any keyword $s \in \mathcal{S}$, it is classified as a non-linear reasoning step; otherwise, it is considered a linear reasoning step. The keyword set $\mathcal{S}$ includes:{*Wait, Alternatively*, *Let me verify*, *another solution*, *Let me make sure*, *hold on*, *think again*, *think differenly*, *another approach*, *another method*}.

## B.2    TRAINING DETAILS FOR LINEAR PROBING

To optimize the linear probe, we first randomly sample 1,000 features from both linear and non-linear thought steps to mitigate class imbalance, as linear steps significantly outnumber non-linear ones. The dataset is then randomly split into training, validation, and test sets with a ratio of $8 : 1 : 1$. We train the linear probe using the Adam optimizer with an initial learning rate of $1 \times 10^{-3}$, which is decayed following a cosine annealing schedule. The final checkpoint is selected based on the highest validation accuracy.

# C    MORE EXPERIMENT RESULTS

## C.1    PROBING ACCURACY OF REASONING REPRESENTATIONS

We report the probing results of different models in Figure 8, 9, 10, 11, and 12 where we can observe that certain attention heads exhibit higher accuracy, *i.e.*, cognitive heads.

## C.2    MORE RESULTS OF ACTIVATION INTERVENTION

We present additional examples in Figure 13, illustrating the reasoning process when the non-linear reasoning component is either enhanced or reduced. Specifically, enhancing non-linear reasoning leads the model to generate longer reasoning chains (e.g., 84 steps), while reducing it results in shorter chains (e.g., 29 steps), compared to the original 31-step output.

# D    CLARIFICATION OF LLM USAGE

In this work, large language models are employed to refine the writing and to aid in generating code for figure plotting. All generated outputs are thoroughly validated by the authors prior to use.

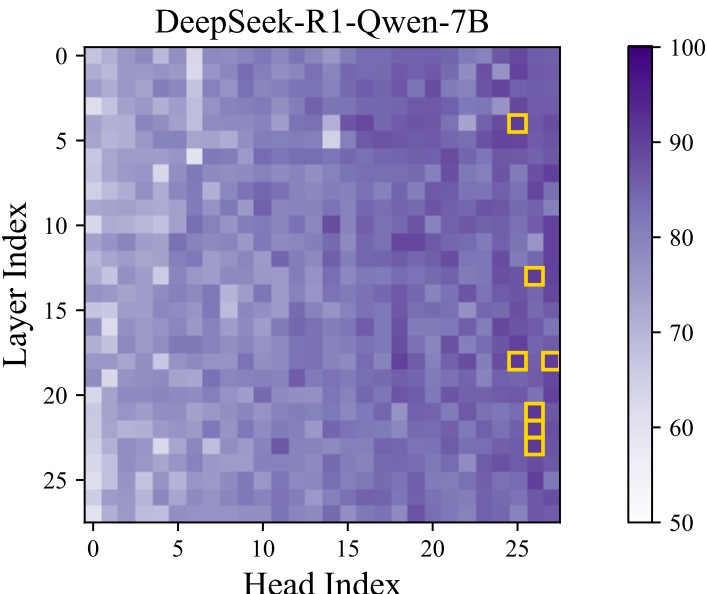

Figure 8: Visualization of probing accuracy for DeepSeek-R1-Distill-Qwen-7B.

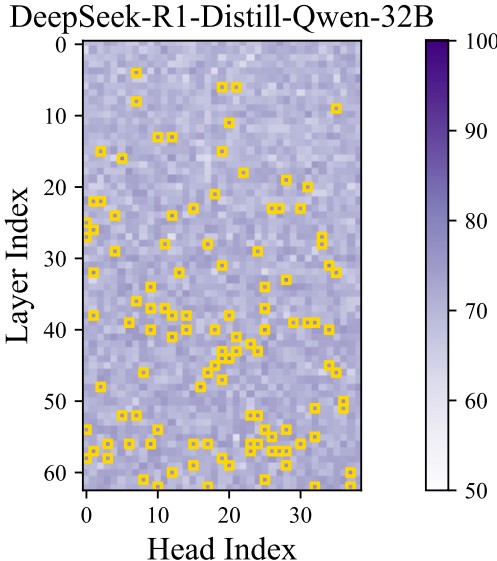

Figure 9: Visualization of probing accuracy for DeepSeek-R1-Distill-Qwen-32B.

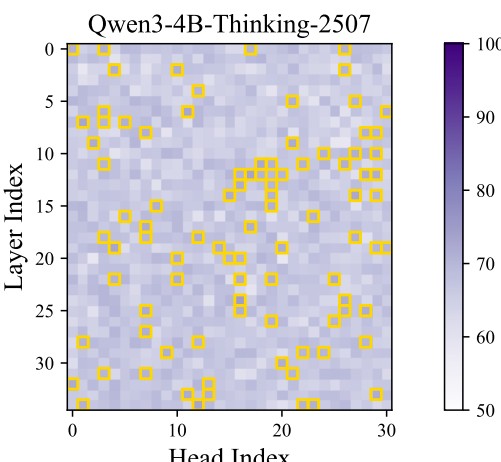

Figure 10: Visualization of probing accuracy for Qwen3-4B.

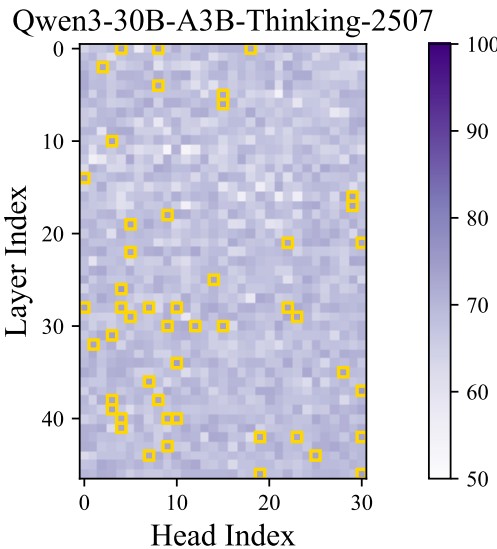

Figure 11: Visualization of probing accuracy for Qwen3-30B.

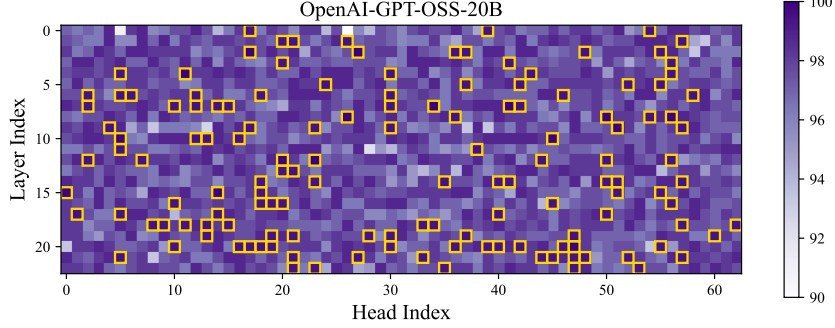

Figure 12: Visualization of probing accuracy for GPT-OSS-20B.

**Question:** What is the smallest positive perfect cube that can be written as the sum of three consecutive integers?

### Enhance Non-Linear Reasoning: 84 Steps

Alright, so I have this problem: find the smallest positive perfect cube that can be written as the sum of three consecutive integers. Hmm, okay. Let me try to figure this out step by step. First, I know that a perfect cube is a number that is an integer raised to the power of three. So, numbers like 1, 8, 27, 64, etc., are perfect cubes. The problem is asking for the smallest one that can be expressed as the sum of three consecutive integers. Okay, let's denote the three consecutive integers. If I let the first integer be $n$, then the next two would be $n + 1$ and $n + 2$. So, the sum of these three numbers is $n + (n + 1) + (n + 2)$. Let me compute that sum: $n + n + 1 + n + 2 = 3n + 3$. So, the sum simplifies to $3n + 3$. Now, we want this sum to be a perfect cube. Let me denote the perfect cube as $k^3$, where $k$ is a positive integer. So, we have the equation:

$$3n + 3 = k^3$$

Hmm, okay. So, I need to solve for $n$ in terms of $k$. Let's rearrange the equation:

$$3n = k^3 - 3$$

$$n = \frac{k^3 - 3}{3}$$

Since $n$ has to be an integer (because we are talking about integers), the numerator $k^3 - 3$ must be divisible by 3. Let's see when that happens. Looking at $k^3 \mod 3$, I remember that for any integer $k$, $k^3 \mod 3$ is equal to $k \mod 3$. Because: - If $k \equiv 0 \mod 3$, then $k^3 \equiv 0 \mod 3$.
- If $k \equiv 1 \mod 3$, then $k^3 \equiv 1 \mod 3$.
- If $k \equiv 2 \mod 3$, then $k^3 \equiv 8 \equiv 2 \mod 3$.
So, $k^3 \equiv k \mod 3$. Therefore, $k^3 - 3 \equiv k - 0 \mod 3$, since 3 is divisible by 3. Wait, no, 3 mod 3 is 0, so $k^3 - 3 \equiv k \mod 3$. So, for $k^3 - 3$ to be divisible by 3, $k$ must be congruent to 0 mod 3.
That is, $k$ must be a multiple of 3. So, let me set $k = 3m$, where $m$ is a positive integer. Substituting back into the equation:

$$n = \frac{(3m)^3 - 3}{3}$$

$$n = \frac{27m^3 - 3}{3}$$

$$n = 9m^3 - 1$$

So, $n$ must be equal to $9m^3 - 1$ for some integer $m \geq 1$. But since we are looking for the smallest positive perfect cube, we should try the smallest values of $m$ and see what $k$ becomes, and then compute $k^3$ to find the smallest one. Let's start with $m = 1$: - $k = 3(1) = 3$
- $k^3 = 27$
- $n = 9(1)^3 - 1 = 8$ So, if $n = 8$, the three consecutive integers are 8, 9, and 10. Let me check their sum: $8 + 9 + 10 = 27$, which is indeed $3^3$. So, 27 is a perfect cube that can be written as the sum of three consecutive integers. **Wait, but hold on a second.** Is 27 the smallest such cube? Let me check with $m = 0$. But $m = 0$ would give $k = 0$, and $k^3 = 0$. But the problem specifies positive integers, so 0 is not positive. So, $m = 1$ is the smallest possible. But just to make sure, let me check if there's a smaller cube with $m$ being a fraction or something, but since $m$ has to be an integer because $k = 3m$, $m$ must be at least 1. So, 27 is the smallest. But wait, hold on again. Is 27 the smallest positive cube that can be expressed as the sum of three consecutive integers? Let me think if there are any smaller cubes that might fit. Wait, the next smaller cube after 1 is 8, which is $2^3$. **Let me see if 8 can be expressed as the sum of three consecutive integers.** Trying to solve $3n + 3 = 8$, so $3n = 5$, $n = 5/3$, which is not an integer. So, 8 cannot be expressed as such. Next cube is 1, but 1 is too small. Let's see, $3n + 3 = 1$, so $3n = -2$, $n = -2/3$. That's also not an integer. So, 1 is not possible. Then, 8 is the next, but as we saw, that doesn't work. So, the next cube is 27, which works. So, 27 is the smallest. Wait, but let me think differently. Maybe I made a mistake in assuming $k$ must be a multiple of 3. **Let me re-examine that step.** We had $3n + 3 = k^3$, so $3(n + 1) = k^3$. Therefore, 3 divides $k^3$, which implies that 3 divides $k$. So, yes, $k$ must be a multiple of 3, so $k = 3m$. So, that part is correct. Therefore, the smallest cube is 27. Wait, but hold on, another thought. Maybe I can approach this problem differently. Let me let the three consecutive integers be $x - 1$, $x$, and $x + 1$. So, their sum is $(x - 1) + x + (x + 1) = 3x$. So, the sum is $3x$, which is a perfect cube. **Ah! So, this is another way to express the problem.** The sum is $3x$, and that needs to be a perfect cube. So, $3x = k^3$, so $x = \frac{k^3}{3}$. So, $k^3$ must be divisible by 3, which again implies that $k$ is divisible by 3, so $k = 3m$, leading to $x = 9m^3$. Therefore, $x$ must be $9m^3$, and the three consecutive integers are $9m^3 - 1$, $9m^3$, and $9m^3 + 1$. So, the smallest such cube is when $m = 1$, so $x = 9 \times 1^3 = 9$. Therefore, the three consecutive integers are 8, 9, 10, and their sum is 27, which is $3^3$. So, same result. Therefore, 27 is indeed the smallest positive perfect cube that can be expressed as the sum of three consecutive integers. Wait, but just to make sure, let me check $m = 1$ gives us 27, which is correct. Let me try $m = 2$: $x = 9 \times 8 = 72$. So, the integers are 71, 72, 73. Their sum is 216, which is $6^3$. So, 216 is another cube, but it's larger than 27, so 27 is still the smallest. Alternatively, if I set $m = 0$, $x = 0$, but that's not positive. So, yeah, 27 is the smallest. **Wait, but hold on. Let me think if I can have a smaller cube, maybe 0, but no, the question specifies positive integers, so 0 isn't positive. So, 27 is the smallest. Wait, but hold on. Let me double-check my initial approach.** When I started, I considered the sum as $3n + 3$, and set that equal to $k^3$. Then, I concluded that $k$ must be a multiple of 3, so $k = 3m$. Then, $n = 9m^3 - 1$. So, for $m = 1$, $n = 8$, and the sum is 27, which is correct. Is there a way to have a smaller cube? Let me see. Wait, suppose I didn't assume $k$ is a multiple of 3, but just let $k$ be any integer. So, for the equation $3n + 3 = k^3$, can $k$ be something else? Wait, earlier I thought that since $3n + 3$ is a multiple of 3, $k^3$ must be a multiple of 3, which implies $k$ is a multiple of 3. So, that seems necessary. So, no, $k$ can't be anything else. Therefore, the smallest $k$ is 3, giving $k^3 = 27$, which is the smallest possible. Alternatively, if I didn't do that substitution, can I get a smaller cube? Let me suppose that $k$ is not a multiple of 3, but then $3n + 3 = k^3$ must still hold. Let's see if there's a case where $k$ is not divisible by 3, but $k^3$ is divisible by 3. But, actually, $k^3$ is divisible by 3 only if $k$ is divisible by 3. Because 3 is prime, so if a prime divides a cube, it must divide the base. So, yes, $k$ must be divisible by 3. Therefore, $k$ must be a multiple of 3, so the smallest $k$ is 3, leading to $k^3 = 27$. Wait, but just to make sure, let me think about other representations. Alternatively, let me think of the sum of three consecutive integers. If I take three consecutive integers, the sum is 3 times the middle number. So, the sum is 3 times the middle term. So, 3 times some integer is a perfect cube. Therefore, $3n = k^3$, so $n = k^3 / 3$. For $n$ to be integer, $k^3$ must be divisible by 3, so $k$ must be divisible by 3. Thus, the smallest $k$ is 3, giving $n = 9$, so the numbers are 8, 9, 10, sum 27. So, same result. Final Answer The smallest positive perfect cube is \boxed{27}. </think>

### Original Reasoning Process: 31 steps

Alright, so I have this problem: find the smallest positive perfect cube that can be written as the sum of three consecutive integers. Hmm, okay. Let me try to figure this out step by step. First, I know that a perfect cube is a number that is an integer raised to the power of three. So, numbers like 1, 8, 27, 64, etc., are perfect cubes. The problem is asking for the smallest one that can be expressed as the sum of three consecutive integers. Let me denote the three consecutive integers. If I let the middle integer be $n$, then the three integers would be $(n - 1)$, $n$, and $n + 1$. Let me compute that sum: $(n - 1) + n + (n + 1)$. Combining like terms, that's $n - 1 + n + n + 1$. The -1 and +1 cancel out, so we have $3n$. Wait, so the sum of three consecutive integers is just three times the middle integer. That's interesting. So, the sum is $3n$. But the problem says the sum has to be a perfect cube. So, $3n$ must be a perfect cube. Let me write that as $3n = k^3$, where $k$ is some positive integer. So, $n = \frac{k^3}{3}$. Since $n$ has to be an integer, $k^3$ must be divisible by 3. Therefore, $k$ must be a multiple of 3, right? Because if 3 divides $k^3$, then 3 must divide $k$. So, let me write $k$ as $3m$, where $m$ is a positive integer. Then, $k^3 = (3m)^3 = 27m^3$. Plugging this back into the equation for $n$, we get $n = \frac{27m^3}{3} = 9m^3$. So, $n = 9m^3$. That means the three consecutive integers are $9m^3 - 1$, $9m^3$, and $9m^3 + 1$. Now, the sum of these three numbers is $27m^3$, which is a perfect cube. So, for each positive integer $m$, we can get a perfect cube as the sum. But the question asks for the smallest positive perfect cube. So, I need to find the smallest $m$ such that $27m^3$ is positive. Well, $m$ is a positive integer, so the smallest $m$ is 1. Plugging $m = 1$ into $27m^3$, we get $27 \times 1^3 = 27$. So, 27 is the smallest positive perfect cube that can be written as the sum of three consecutive integers. Wait, let me check if that's correct. If $m = 1$, then the three consecutive integers are $9(1)^3 - 1 = 8$, $9(1)^3 = 9$, and $9(1)^3 + 1 = 10$. So, adding them together: $8 + 9 + 10 = 27$. Yep, that works. But just to make sure I didn't make a mistake earlier, let me think again. So, if the sum is $3n$, and $3n$ must be a perfect cube, then $n$ must be a multiple of 3 because 3 divides into it. So, $n$ is $3m$, which leads to the sum being $27m^3$. So, yeah, 27 is the smallest one. Is there a smaller perfect cube that can be written as the sum of three consecutive integers? Well, the perfect cubes are 1, 8, 27, 64, etc. Let's see if I can written as such a sum. If $3n = 1$, then $n = 1/3$, which isn't an integer, so that doesn't work. Next is 8. So, $3n = 8$, n is $8/3$, which isn't an integer either. Then, 27. As I found earlier, 27 can be written as $8 + 9 + 10$. So, 27 works. Therefore, 27 is indeed the smallest. Wait, but hold on, could 1 be considered? But 1 is too small because $3n = 1$ would require n to be 1/3, which is not an integer. So, 1 is out. 8 is also too small because $3n = 8$ would need n = 8/3, which is not an integer. So, 8 is also out. So, the next one is 27. Therefore, 27 is the smallest. STEP:17##### Alternatively, maybe I can think of it differently. Suppose the three consecutive integers are $a, a + 1, a + 2$. Their sum is $3a + 3 = 3(a + 1)$. So, $3(a + 1)$ must be a perfect cube. Therefore, $a + 1$ must be a perfect cube divided by 3. So, $a + 1 = k^3/3$. So, $k^3$ must be divisible by 3, so $k$ must be a multiple of 3, as before. .... Final Answer The smallest positive perfect cube that can be written as the sum of three consecutive integers is \boxed{27}. </think>

### Reduce Non-Linear ReasoningL 29 steps

Alright, so I have this problem: find the smallest positive perfect cube that can be written as the sum of three consecutive integers. Hmm, okay. Let me try to figure this out step by step. First, I know that a perfect cube is a number that is an integer raised to the power of three. So, numbers like 1, 8, 27, etc., are perfect cubes. The problem is asking for the smallest one that can be expressed as the sum of three consecutive integers. Let me denote the three consecutive integers. If I let the middle integer be $n$, then the three integers would be $n - 1$, $n$, and $n + 1$. So, their sum would be $(n - 1) + n + (n + 1)$. Let me compute that sum: $(n - 1) + n + (n + 1)$. Combining like terms, that's $n - 1 + n + n + 1$. The -1 and +1 cancel out, so we have $3n$. So, the sum of three consecutive integers is $3n$. Now, the problem states that this sum is a perfect cube. So, $3n$ must be a perfect cube. Let me write that as an equation: $3n = k^3$, where $k$ is some positive integer. So, $n = \frac{k^3}{3}$. Since $n$ must be an integer (because we're dealing with integers), $k^3$ must be divisible by 3. That means $k$ itself must be divisible by 3, because 3 is a prime number. So, let me let $k = 3m$, where $m$ is an integer. Substituting back into $n$, we get $n = \frac{(3m)^3}{3} = \frac{27m^3}{3} = 9m^3$. So, $n = 9m^3$. Therefore, the three consecutive integers are $n - 1 = 9m^3 - 1$, $n = 9m^3$, and $n + 1 = 9m^3 + 1$. Now, let's compute the sum again: $(9m^3 - 1) + 9m^3 + (9m^3 + 1) = 27m^3$. Wait, that's interesting. The sum is $27m^3$, which is $(3m)^3$, confirming that it is indeed a perfect cube. But hold on, the problem is asking for the smallest positive perfect cube. So, I need to find the smallest $k$ such that $k^3 = 3n$, with $n$ being an integer. From the earlier step, $k = 3m$, so the smallest positive $k$ is 3, because $m$ has to be a positive integer as well. Let me check that. If $m = 1$, then $k = 3$, so $k^3 = 27$. Then $n = 9m^3 = 9(1)^3 = 9$. So, the three consecutive integers are 8, 9, 10. Let's add them up: $8 + 9 + 10 = 27$. Perfect, that works. ... Final Answer The smallest positive perfect cube that can be written as the sum of three consecutive integers is \boxed{27}. </think>

