# OpenReview forum: "Understanding and Steering the Cognitive Behaviors of Reasoning Models at Test-Time"
_ICLR.cc/2026/Conference — ICLR 2026 Conference Desk Rejected Submission_

### Official Review · Reviewer_PbFD · 2025-11-01

**Soundness:** 3
**Presentation:** 3
**Contribution:** 2
**Rating:** 4
**Confidence:** 3

**Summary:**

The paper investigates how large reasoning models exhibit and internalize cognitive-like behaviors such as verification and backtracking.
The new approach is identify a subset of “cognitive attention heads” whose activations correlate with these behaviors and propose CREST, a training-free, test-time steering method that modifies head-level activations to suppress or enhance such behaviors. Empirically, CREST improves accuracy and reduces token usage across multiple reasoning benchmarks.

**Strengths:**

S1. The proposed method is effective. It achieves consistent accuracy gains (up to +17.5%) while reducing generation length (up to −37.6%) on several reasoning benchmarks and models.

S2. The discovery of “cognitive heads,” the probing analysis, and activation visualizations provide interpretable insights into internal reasoning mechanisms.

S3. CREST requires no retraining or gradient updates, can be applied across many LLMs, and introduces negligible inference overhead.

**Weaknesses:**

**W1**. The update $\hat{x_{i,j}}=x_{i,j}-\alpha v_{i,j}$ and its norm-preserving variant lack theoretical justification. It remains unclear why subtracting or orthogonalizing along an empirical mean direction improves reasoning, or how this compares to simpler scaling methods. In addition, the novelty of this intervention is unclear: please position it against existing activation/attention-head steering techniques (e.g., activation engineering / latent-direction editing, manifold steering for overthinking mitigation, and SEAL’s task-arithmetic latent calibration), and clarify whether a head-level, per-step, norm-preserving projection is genuinely new relative to those practices.

**W2**. The head selection uses reasoning-type labels rather than accuracy supervision, yet the paper attributes final accuracy gains to “better reasoning.” The mechanism connecting suppression of non-linear reasoning and higher task accuracy is not rigorously analyzed. In particular, non-linear reasoning (verification/backtracking) can be crucial for error correction and indiscriminate suppression might harm exploration ability. A simple test: Reporting Pass@k accuracy or measuring recovery after early mistakes, which could reveal whether CREST causes “collapse of entropy.”

**W3**. Steps are delimited by the "\n\n" boundary and non-linear steps are defined by keyword matching (e.g., Wait, Alternatively). This is heuristic and may entangle style/semantics with “cognitive type”. The paper does not validate these labels against model-centric signals (e.g., entropy/perplexity, confidence slope). Thus it is unclear whether the labels faithfully capture reasoning modes rather than surface cues. Moreover, keyword usage and even the choice of step boundaries can vary across models, so the generality of keyword-based labeling is uncertain. I encourage the authors to evaluate on a broader set of models and to report the distributions of (i) “\n\n” step boundaries and (ii) non-reasoning keywords across different datasets and models, to substantiate the cross-model applicability of the keyword-matching scheme.

**Questions:**

See weakness.

---

> ### Author Response · Authors · 2025-11-26
> **Responses to Reviewer PbFD [Q1-Q3]**
>
> We appreciate that Reviewer PbFD acknowledges the insights from our analysis of cognitive heads and recognizes the effectiveness and consistency of our method. The remaining comments are highly constructive, and we address them point by point below.
>
>
> **[Q1: Theoretical Justification.]** Thanks for the question. The theoretical foundation of our method builds on the linear representation hypothesis [1], which suggests that in well-trained LLMs, the representation space is highly structured and contains linear directions corresponding to specific semantic concepts. A classic example is the analogy “King − Man + Woman = Queen” [2]. This implies that both linear and nonlinear reasoning patterns may also correspond to identifiable linear directions in the activation space—an insight we empirically verify in our work. This is why activation arithmetic can effectively steer reasoning behavior.
>
> We fully acknowledge the importance of deeper theoretical justification. However, formal proofs for neural networks remain extremely challenging, and existing theoretical results typically focus on small models with one or two layers and often omit position encoding [3]. In contrast, our reasoning setting involves long-context interactions and additional complexity stemming from modern large-scale models. As such, a complete theoretical treatment is beyond the scope of this work. We plan to explore this direction in future work, while incorporating insights from the linear representation hypothesis to further clarify the intuition behind our method.
>
> Regarding the comparison with simpler scaling approaches, we would like to clarify that our method is orthogonal to scaling. Scaling improves accuracy by increasing inference cost, whereas our work focuses on optimizing the accuracy–efficiency trade-off within a single inference run. Our approach can also be directly applied in scaling scenarios. As an initial investigation in Table R1, our method continues to provide benefits across multiple inference runs, achieving higher pass@k with lower token cost.
>
>
> **[Q2: Clarification of Novelty]** Thanks. First, our work provides new insights into the reasoning process: we identify that linear and nonlinear reasoning behaviors are both detectable and exhibit head-specific properties. These findings relate to prior studies on head-level specialization in long-context modeling [7–8], and they contribute to a deeper understanding of reasoning mechanisms that is beneficial for the community. This insight has not been discussed in prior practice [4–6], which is why we believe our work offers novel conceptual contributions—as also noted by Reviewer AvFd, who described our results as “a finding that deepens model interpretability.”
>
> Second, on the methodological side, both our head-level design and norm-preserving projection are new relative to prior work [4–6]. While [6] also uses a per-step steering strategy, our method is directly motivated by our head-wise analysis, allowing for more precise and targeted steering. This design choice leads to consistent improvements over SEAL in our experiments, supporting the technical novelty of our method.
>
>
> **[Q3: Pass@k analysis of CREST]** Great suggestion. We agree that evaluating Pass@k performance is important to assess the exploration capability of CREST. We report Pass@k results for R1-1.5B on Math-500 in Table R1. Across all values of k, CREST consistently improves over the baseline, yielding accuracy gains of 0.2–0.8% while substantially reducing token usage. These results indicate that CREST remains effective under the Pass@k evaluation protocol and does not compromise the model’s ability to explore diverse reasoning trajectories.
>
> Furthermore, inspired by recent findings [9], activation–based interventions offer a principled mechanism for modulating the model’s reasoning process. Such modulation can be applied with varying strengths or along different intervention directions, thereby enabling additional diversity when needed. This plug-in diversity further improves accuracy in multi-run settings, further supporting the superiority of CREST for reasoning improvement.
>
> Table R1. Comparison of Pass@k Results for R1-1.5B on Math-500
>
> | Pass@k            | Baseline Accuracy | Baseline Token-Length | CREST Accuracy | CREST Token-Length |
> | ----------------- | ---------------------- | -------------------------- | ------------------- | ----------------------- |
> | **Pass@1**  | 84.0                  | 5497.0                     | 84.8               | 4106.7                  |
> | **Pass@4**  | 92.8                   | 5099.0                     | 93.2                | 3917.1                  |
> | **Pass@8**  | 94.6                   | 5122.4                     | 94.8                | 3894.2                  |
> | **Pass@16** | 96.0                   | 5117.6                     | 96.4                | 3875.9                  |

---

> ### Author Response · Authors · 2025-11-26
> **Responses to Reviewer PbFD[Q4]**
>
> **[Q4: Robustness of Keyword Matching Method.]** Thank you for the question. First, we would like to clarify that our method uses the MATH500 training set as a calibration set to identify steering vectors. These vectors are then evaluated on tasks beyond the original training distribution. As shown in Tables 1 and 3, the identified vectors generalize well across both intra-domain and cross-domain reasoning tasks. This supports the robustness of our keyword-based method in capturing transferable reasoning behaviors across domains.
>
> Additionally, we conducted a further validation of the keyword-based annotations by comparing them against GPT-4o-generated labels. On the MATH500 benchmark, steering using LLM-generated labels achieves comparable performance to keyword-based labels—85.00 vs. 84.80 accuracy with R1-1.5B. Moreover, the overlap between these two annotation approaches is 86%, indicating that our method captures deeper reasoning patterns beyond surface-level textual cues. These results further support the reliability of our keyword-based approach.
>
>
> [1] The Linear Representation Hypothesis and the Geometry of Large Language Models.
>
> [2] https://kawine.github.io/blog/nlp/2019/06/21/word-analogies.html.
>
> [3] Scan and Snap: Understanding Training Dynamics and Token Composition in 1-layer Transformer.
>
> [4] Steering Language Models with Activation Engineering.
>
> [5] Mitigating Overthinking in Large Reasoning Models via Manifold Steering.
>
> [6] SEAL: Steerable Reasoning Calibration of Large Language Models for Free.
>
> [7] Retrieval Head Mechanistically Explains Long-Context Factuality.
>
> [8] DuoAttention: Efficient Long-Context LLM Inference with Retrieval and Streaming Heads.
>
> [9] Fractional Reasoning via Latent Steering Vectors Improves Inference Time Compute.

---

> ### Comment · Reviewer_PbFD · 2025-11-27
>
> The authors have addressed most of my earlier concerns. I am generally satisfied with the clarifications and am willing to raise my score. Before finalizing it, there is one point I still hope the authors can clarify:
>
> 1. Token-Length in Table R1 (score-relevant).
>
> I am unsure how Token-Length is defined when reporting Pass@k. In Table R1, CREST’s Token-Length decreases as k increases, while the baseline does not show the same trend. I would greatly appreciate a clear explanation of how Token-Length is computed in this setting.
>
> ---
>
> The next two points are more out of personal curiosity:
>
> 2. Design process of the online intervention (Function 3).
>
> I notice Figure 4 shows that the intervention works well across a range of $\alpha$. I’m simply curious whether Function (3) was the first formulation you tried, or if other variants were explored before settling on the current design.
>
> 3. Possibility of adaptive intervention.
>
> Inspired by Figure 4 and reference [9], I wonder whether an **adaptive** approach might be possible. For example, allowing $\alpha$ to become negative when the model is uncertain (to encourage exploration), and increasing α when the model becomes confident, e.g. measured by entropy. This is not a request for additional experiments; I’m simply interested in whether the authors think such an adaptive mechanism is realistic. If something like this actually works, I think it worths of a much higher score (8+).

---

> ### Author Response · Authors · 2025-11-28
> **Responses to the follow-up questions**
>
> Thanks for the responses and raising these insightful questions. Point-wise responses are below.
>
> **Token Length**: First, we would like to clarify the averaging process used in Table R1. The token length is first averaged across samples and then across individual runs. Due to an implementation issue, as k increases, only the samples that remain incorrect are included in the calculation. This inadvertently places greater weight on more difficult examples in later steps. Typically, for a given sample across multiple runs, token lengths are shorter when the model prediction is correct compared to when it is incorrect. (In our case, the average token length for correct vs. incorrect predictions is 5191 vs. 10732 in CREST, and 5947 vs. 16159 in the Baseline.) This explains the observed trend of decreasing token length as k increases. As shown in Table R1, both the Baseline and CREST exhibit this general downward trend and eventual convergence.
>
> To further examine token length behavior, we report two additional analyses: (i) the mean and standard deviation of token lengths across 16 individual runs: 4151 ± 141 for CREST and 5320 ± 205 for the Baseline; and (ii) the average token length computed directly across all samples and runs. In this setting, runs are randomly sampled as k increases. The updated results, presented in Table R2, demonstrate our earlier findings: CREST delivers more stable and consistent improvements under the pass@k setting.
>
> Table R2. Comparison of Pass@k Results for R1-1.5B on Math-500
>
> | Pass@k            | Baseline Accuracy | Baseline Token-Length | CREST Accuracy | CREST Token-Length |
> | ----------------- | ---------------------- | -------------------------- | ------------------- | ----------------------- |
> | **Pass@1**  | 84.2                  |5101.6                   | 84.8               | 4106.7                  |
> | **Pass@4**  | 93.4                   |5283.4                     | 93.6                | 4071.3                  |
> | **Pass@8**  | 95.0                   | 5316.9                    | 95.8                | 4169.9                  |
> | **Pass@16** | 96.0                   | 5320.1                    | 96.4                | 4151.3                  |
>
> **Design Process of the Online Intervention (Function 3)**: Good catch. Yes, we initially implemented the steering method as Function 3. However, its main drawback was the introduction of an additional hyperparameter, which required careful tuning across different models and tasks. To address this, we transitioned to the current norm-preserving approach, which offers greater stability across tasks and eliminates the need for tuning the coefficient $\alpha$.
>
> **Possibility of Adaptive Intervention:** We believe adaptive intervention is both realistic and promising, as suggested by reference [9], where varying the steering coefficient $\alpha$ across runs has been shown to improve diversity and, in turn, enhance pass@k performance. Broadly, this problem can be decomposed into two sub-problems: (i) Adaptive steering within a single run, and (ii) Adaptive steering across different runs. For (i), the key challenge lies in identifying when the model is engaging in unnecessary reasoning versus when it is confident. In uncertain segments, we can intervene more aggressively in the latent space. Achieving this requires step-level confidence estimation, which could be guided by intrinsic model signals such as entropy, or via supervised methods—e.g., training an auxiliary predictor on contrastive chain-of-thought datasets to identify high/low confidence steps. For (ii), one strategy is to apply stronger steering early on to enforce more deterministic reasoning modes—helping eliminate redundant or unhelpful non-linear behaviors. Then, for samples where responses converge across runs (indicating low diversity), we can gradually reduce or even reverse the steering strength to encourage more varied, non-linear reasoning trajectories. We view these directions as promising avenues for future work to further refine and enhance the reasoning process.

---

### Official Review · Reviewer_AvFd · 2025-11-02

**Soundness:** 3
**Presentation:** 2
**Contribution:** 3
**Rating:** 6
**Confidence:** 4

**Summary:**

This paper introduces CREST, a method for understanding and steering the cognitive behaviors of large language models (LLMs) during test-time reasoning. The central idea is that specific attention heads, termed “cognitive heads”, exhibit activation patterns that correlate with linear and non-linear (e.g., verification, backtracking) reasoning steps. The authors propose a training-free, test-time intervention that identifies cognitive heads using linear probes and constructs steering vectors from their activations; applying vector-based interventions at inference time suppresses or amplifies certain cognitive behaviors. Experiments demonstrate that across several tasks and models, CREST improves accuracy (up to 17.5%) while substantially reducing token usage (up to 37.6%). The method generalizes across domains and models without retraining or heavy hyperparameter tuning.

**Strengths:**

1. The paper comprehensively demonstrates, both quantitatively (see Figure 1, Figure 8-12) and qualitatively, that certain attention heads are strongly predictive of cognitive behaviors such as linear vs. non-linear reasoning steps, a finding that deepens model interpretability.
2. CREST requires only offline calibration (linear probing plus simple covariance projection), imposing negligible runtime overhead and no further model training. Intervention is mathematically straightforward (see Equation in Section 4.2) and does not depend on sensitive hyperparameter tuning.
3. The method is benchmarked on math, code, planning, and common-sense tasks (Table 1–3), along with extensive ablations (Figure 6, Figure 7) over head selection and response length distributions.

**Weaknesses:**

1. While the probing accuracies and covariance analyses are compelling, the theoretical motivation for why cognitive heads manifest at specific layers and how robust these interventions are (especially in models with cross-layer dependencies or strong MoE gating) remains thin. Theoretical claims about head specialization rely mainly on correlational evidence (Figure 1, Figure 8-12) instead of providing more causal or mechanistic arguments. Section 4.1.2 describes projecting to a low-rank subspace, but the rationale for working at the layer-aggregated level (beyond noise reduction) is not well justified.
2. The procedure for labeling steps as linear/non-linear depends on keyword matching with a fixed list (Appendix B.1), potentially introducing false positives/negatives. No quantitative sensitivity or robustness analysis for this heuristic is provided.
3. The paper demonstrates average improvements across benchmarks but does not provide in-depth analyses, e.g., on when steering harms reasoning fidelity, or on failure cases tied to specific reasoning motifs. Figure 7 hints at failure cases reaching token limits, but beyond noting repetition, there is no nuanced discussion.
4. Figure 3 and additional qualitative example in supplement (Figure 13) offer single-case or anecdotal illustrations; a more systematic qualitative study (e.g., did steering consistently reduce unnecessary backtracking or, conversely, miss subtle forms of needed verification?) would support generality claims.

**Questions:**

See weakness

---

> ### Author Response · Authors · 2025-11-26
> **Responses to Reviewer AvFd [Q1-Q2]**
>
> We thank Reviewer AvFd for the positive feedback and constructive suggestions. Our point-by-point response is provided below.
>
> **[Q1: Theoretical Justification and Clarification.]**
> Thank you for the question. We would like to clarify that we do provide causal evidence of head specialization in Figure 4, where intervening on the identified cognitive heads consistently alters the model’s reasoning style, while intervening on non-cognitive heads produces minimal change. This supports the claim that certain heads play a functional role in reasoning.
> Regarding why these heads tend to appear at specific layers: this observation is consistent with prior work showing that deeper transformer layers encode more abstract and semantic information [1,2]. Developing a full theoretical explanation for head specialization at current model scales remains challenging. Existing theoretical frameworks for transformer interpretability predominantly focus on small models with one/two layers, while modern LLMs introduce substantial additional complexity due to their depth, size, and long-context interactions. Therefore, a formal theoretical treatment is beyond the scope of our work. Our goal in this paper is to provide empirical insights into the reasoning process and propose a training-free, plug-and-play approach to make reasoning both more efficient and more accurate.
> Our rationale for using layer-aggregated representations stems from two considerations. First, each attention head operates in its own representation space, but the model’s forward pass aggregates these spaces, producing a highly compatible integrated representation. Performing low-rank projection independently at the head level leads to risk of incompatible subspaces across heads. Empirically, we find that head-wise projections make performance highly sensitive and results fluctuate with the number of chosen heads. This sensitivity makes a head-wise approach less practical due to its reliance on unstable hyperparameters turning. In contrast, aggregating representations across layers yields a stable shared subspace and more consistent performance and hyperparameter selection as shown in Figure 6.
>
> **[Q2: Robustness of Keyword Matching Method.]** Good point. We acknowledge that using keyword-based labeling may introduce some degree of noise. That said, our method is designed to be robust to such noise, supported by the following: (i) Noise resilience through averaging: The steering direction is computed by averaging activation vectors from multiple samples exhibiting the target behavior. This averaging process naturally dampens the effect of a small number of mislabeled examples, making the overall direction stable. (ii) Low-rank projection for denoising: As described in Section 4.1.2, we further refine the direction via low-rank projection, which helps to extract the dominant behavioral subspace and reduce the impact of noise or outliers in the representation. (iii) Validation with LLM-based annotations: To further assess robustness, we repeated the experiment using behavior labels generated by GPT-4o instead of keyword matching. The resulting steering vector yielded similar performance on the MATH-500 benchmark, i.e., 84.80 (keyword) vs. 85.00 (LLM-based) with R1-1.5B, confirming that our approach is not sensitive to the specific labeling mechanism.

---

> ### Author Response · Authors · 2025-11-26
> **Responses to Reviewer AvFd [Q3-Q4]**
>
> **[Q3: Analysis of Failure Cases]** Thank you for the insightful comment. We acknowledge that CREST is not without limitations. Beyond the repetition cases mentioned in Figure 7, we observe additional failure modes. Notably, in some instances, CREST outputs deviate from strict adherence to the task instructions, for example, failing to place the final answer within the required format (e.g., missing a boxed answer). Such deviations can introduce noise during output parsing, affecting automatic evaluation.
>
> Furthermore, Table R1 highlights that most of the performance gains occur on easier questions. This suggests that CREST primarily helps prune redundant or “overthinking” reasoning steps, rather than extending the model’s reasoning capability for more difficult tasks. On hard questions, the accuracy remains comparable to the baseline, indicating CREST’s impact is more aligned with improving conciseness and reducing unnecessary cognitive detours, rather than enabling fundamentally stronger reasoning.
>
>
>
> Table R1: Comparison of reasoning steps across different difficulty levels.
>
> | Difficulty Level | Baseline Linear | Baseline Non-Linear | Baseline Tokens | Baseline Accuracy | CREST Linear | CREST Non-Linear | CREST Tokens | CREST Accuracy |
> |------------------|----------------:|---------------------:|----------------:|------------------:|-------------:|-----------------:|-------------:|---------------:|
> | 1                |          37.09  |               29.16  |         2787.12 |             0.88  |        33.84 |            6.35  |      1646.09 |           0.95 |
> | 2                |          54.57  |               20.90  |         2725.44 |             0.92  |        44.29 |            6.27  |      1900.47 |           0.92 |
> | 3                |          79.94  |               26.88  |         4307.02 |             0.94  |        65.19 |            8.00  |      2608.56 |           0.94 |
> | 4                |         158.30  |               30.39  |         5426.41 |             0.83  |       114.16 |           10.16  |      4046.09 |           0.82 |
> | 5                |         216.03  |               67.92  |         8845.00 |             0.72  |       212.26 |           18.94  |      7607.31 |           0.72 |
>
> **[Q4: Systematic Study on the Effect of CREST]** Thank you for pointing this out! We would like to clarify that we do include a quantitative analysis in Figure 4 in addition to qualitative examples. In Figure 4, we show that increasing the steering strength consistently reduces the number of nonlinear reasoning steps, providing systematic evidence that supports our overall claims.
>
> [1] Fantastic semantics and where to find them: Investigating which layers of generative llms reflect lexical semantics.
>
> [2] Exploring concept depth: How large language models acquire knowledge and concept at different layers?

---

### Official Review · Reviewer_wvoR · 2025-11-03

**Soundness:** 3
**Presentation:** 3
**Contribution:** 2
**Rating:** 4
**Confidence:** 4

**Summary:**

This paper addresses the imbalance between “overthinking” and “underthinking” of reasoning models. They investigate the internal cognitive structure of large language models and identify specific attention heads that correlate with reasoning behaviors such as verification and backtracking. Building on these insights, they propose CREST, which uses linear probing to detect “cognitive heads” and modulates their hidden activations to improve reasoning efficiency. Experiments across multiple models and reasoning benchmarks show that CREST enhances accuracy and reduces token generation, demonstrating strong generalization across domains.

**Strengths:**

1. The paper offers convincing empirical evidence that internal attention heads of LLMs encode semantically meaningful cognitive behaviors such as non-linear reasoning, supported by systematic linear probing.
2. The proposed CREST framework is lightweight and effective, requiring only offline calibration on small datasets and operating at inference time with negligible overhead. It is training-free and model-agnostic.
3. The results span a wide range of tasks and model scales, demonstrating strong generalization and robustness.

**Weaknesses:**

1. The annotation of cognitive behaviors is rather coarse. Classifying linear and non-linear reasoning through surface keywords (e.g., “Wait”, “Alternatively”) may introduce noise.
2. Although the vector adjustments in Equations (4) and (5) are reasonable, there is a lack of intuitive explanation for their geometric or semantic impacts, making it difficult for readers to understand how such rotations affect reasoning patterns.
3. It’s unclear when steering might reduce accuracy by skipping key reasoning steps or oversimplifying explanations. Identifying task types where CREST’s steering could harm performance, and clarifying its reliability under domain shifts, would strengthen the analysis.

**Questions:**

1. Section 3.3 mentions that the parameter α can be adjusted to control the enhancement of linear or non-linear reasoning. However, it doesn’t seem to be used in the later parts. Could the authors clarify how the model decides when to enhance or suppress a certain type of reasoning?
2. Could the authors provide additional numbers showing the proportion of linear or non-linear reasoning used for questions of different difficulty levels? This would help demonstrate how the method alleviates underthinking and overthinking.

---

> ### Author Response · Authors · 2025-11-26
> **Responses to Reviewer wvoR [Q1-Q3]**
>
> We’re glad that Reviewer wvoR acknowledged our method as “lightweight/effective” and our empirical evidence as “convincing”. To address Reviewer wvoR’s concerns, we provide detailed responses in the following.
>
> **[Q: Potential Noise from Keyword-Based Labeling]** Thank you for the question. We agree that direct keyword matching may introduce labeling noise. However, we would like to clarify that our steering method is inherently noise-tolerant for several reasons: (i) Averaging mitigates noise: The final steering direction is obtained by averaging over multiple activation vectors associated with the target behavior. As a result, a small number of mislabeled samples have negligible influence on the aggregated direction due to the averaging effect. (ii) Low-rank projection filters noise: In Section 4.1.2, we apply a low-rank projection step to isolate the most salient shared components of the target behavior. This step further helps suppress noisy or outlier directions, improving stability and robustness. (iii) To further address this concern, we conducted an additional experiment in which the steering vector was derived from behavioral annotations generated by GPT-4o. On the MATH-500 benchmark, steering based on these LLM-generated labels achieves comparable performance, i.e., 84.80 (keyword-based) vs. 85.00 (LLM-based) with R1-1.5B, demonstrating that our method is robust to the choice of labeling source.
>
>
>
> **[Q2: Intuitive Explanation for the Impact of Rotations]** Good suggestion. The intuition behind the effect of rotation comes from the linear representation foundation [1]. A well-known example is the analogy “king − man + woman = queen,” which illustrates that certain semantic directions exist in activation space. Real activations can therefore be interpreted as combinations of multiple conceptual vectors. In our method, the rotation operation is used to selectively remove a specific direction in order to reduce redundant reasoning steps. The norm-preserving term ensures activation stability, since without it the activation magnitude may change excessively when the cosine similarity between the activation and the target concept vector is close to one. We will include this explanation in the revised draft.
>
>
> **[Q3: Identifying When CREST Harm Performance]** Thanks for the question. We would like to clarify that our method focuses on improving the average reasoning capability of current models, rather than guarantee improvements for all possible inputs. In our experiments, there are indeed occasional failure cases of CREST.
> However, in Table 3, we evaluate our method across multiple domains and observe consistent improvements, suggesting that these failure cases are not domain-specific.
> Additionally, in our implementation, we treat the number of steering heads as a hyperparameter to control the steering strength and avoid oversimplifying reasoning traces. The results in Figure 6 indicate that using a moderate number of heads helps prevent oversimplification while achieving robust improvements across domains.
> We also agree that more adaptive steering mechanisms could further improve performance on arbitrary inputs, for example, using model-centric signals such as entropy to estimate confidence in reasoning steps and adaptively adjust the steering strength. However, these signals are imperfect and may lead to false positives, and developing a fully reliable version may require additional training or calibration of the model’s reasoning behavior. In this work, we intentionally focus on a training-free, plug-and-play approach for its simplicity and practicality. Exploring more adaptive steering strategies is an exciting direction that we plan to pursue in future work.

---

> ### Author Response · Authors · 2025-11-26
> **Responses to Reviewer wvoR [Q4-Q5]**
>
> **[Q4: Clarification of Steering Strength $\alpha$]** Great chat! In Section 3.3, we introduce a hyperparameter $\alpha$ to control the steering strength and demonstrate the causal effect of manipulating reasoning steps. This allows us to verify that the identified vector is indeed linked to the targeted behavior. However, relying on $\alpha$ introduces sensitivity and task-specific tuning, which can limit practical applicability. Our method addresses this by incorporating a norm-preservation strategy, which stabilizes the steering effect across different tasks and models and reduces dependence on a manually tuned $\alpha$.
> In our experiments, we observe that current models often exhibit redundancy in their reasoning processes, and removing the nonlinear direction helps alleviate this issue. Therefore, our method does not aim to precisely detect and eliminate specific failing reasoning steps. Instead, we introduce a latent-space prior that reduces the model’s tendency to produce redundant reasoning steps overall.
> For a more precise steering approach, we agree this is an interesting direction—for instance, incorporating model-centric heuristics or adding training-based adjustments to better determine when and how much to steer. This, however, goes beyond our current training-free setup due to practical applicability, and we will explore in the future.
>
> **[Q5: Results of linear and non-linear reasoning steps]** Thanks for the question. We report the number of tokens, non-linear reasoning steps, linear steps, and accuracy for both the baseline and our CREST method across questions of varying difficulty in Table R1. The difficulty level is obtained from the definition in the MATH500 dataset. The results show that CREST effectively reduces the number of non-linear reasoning steps across all difficulty levels. Notably, the improvements in accuracy are more pronounced on easier questions, suggesting that CREST can mitigate overthinking thus enhancing the accuracy.
>
> Table R1: Comparison of reasoning steps across different difficulty levels.
>
> | Difficulty Level | Baseline Linear | Baseline Non-Linear | Baseline Tokens | Baseline Accuracy | CREST Linear | CREST Non-Linear | CREST Tokens | CREST Accuracy |
> |------------------|----------------:|---------------------:|----------------:|------------------:|-------------:|-----------------:|-------------:|---------------:|
> | 1                |          37.09  |               29.16  |         2787.12 |             0.88  |        33.84 |            6.35  |      1646.09 |           0.95 |
> | 2                |          54.57  |               20.90  |         2725.44 |             0.92  |        44.29 |            6.27  |      1900.47 |           0.92 |
> | 3                |          79.94  |               26.88  |         4307.02 |             0.94  |        65.19 |            8.00  |      2608.56 |           0.94 |
> | 4                |         158.30  |               30.39  |         5426.41 |             0.83  |       114.16 |           10.16  |      4046.09 |           0.82 |
> | 5                |         216.03  |               67.92  |         8845.00 |             0.72  |       212.26 |           18.94  |      7607.31 |           0.72 |
>
>
> [1] The Linear Representation Hypothesis and the Geometry of Large Language Models.

---

### Note · Program_Chairs · 2026-01-17
**Submission Desk Rejected by Program Chairs**

The following references in this submission do not refer to real documents and/or have major errors in bibliographic information:

 Sophie Toth and James I.D. Campbell. Backtracking and cognitive load in solving the tower of hanoi. Cognitive Psychology, 131:101434, 2022. doi: 10.1016/j.cogpsych.2022.101434.